# Ripple Effect Protocol:
# Coordinating Agent Populations

## Abstract

Modern AI agents can exchange messages using protocols such as A2A and ACP, yet these mechanisms focus on communication rather than coordination. As agent populations grow, this limitation leads to brittle collective behavior, where individually "smart" agents converge on poor group outcomes. We introduce the *Ripple Effect Protocol (REP)*, a coordination protocol in which agents share not only their decisions but also lightweight *sensitivities*—signals that express how their choices would change if key environment variables shifted. These sensitivities ripple through local networks, enabling groups to align faster and more stably than with decision-only communication. We formalize REP's protocol specification, separating required message schemas from optional aggregation rules, and evaluate it across three domains: supply chain information cascades (Beer Game), preference aggregation in sparse networks (Movie Scheduling), and sustainable resource allocation (Fishbanks). Across these experiments, REP consistently improves coordination accuracy and communication efficiency, while flexibly handling both numerical and textual signals from LLM-based agents. By making coordination a protocol-level capability, REP provides scalable infrastructure for the emerging Internet of Agents. The REP SDK will be released with this paper.

## 1 Introduction

The proliferation of LLM-based agents across web services, enterprise environments, and IoT devices is creating a new coordination challenge: how can intelligent agents that express reasoning in natural language coordinate effectively without centralized control? Unlike traditional multi-agent systems designed for controlled simulations, this emerging "Agentic Web" requires coordination protocols that work with heterogeneous agents developed independently by different organizations, each with distinct objectives and reasoning capabilities Yang et al. (2025); Wang et al. (2025).

Existing coordination approaches assume either centralized design (multi-agent reinforcement learning Sunehag et al. (2017); Rashid et al. (2018)), shared global objectives (consensus algorithms) Ongaro & Ousterhout (2014); Castro & Liskov (1999), or orchestrated control (LLM coordination frameworks like MetaGPT Hong et al. (2023) and AgentVerse Chen et al. (2023)). These paradigms work within controlled environments but cannot scale to independently owned agents operating across trust boundaries. Modern agent communication protocols like A2A Google for Developers (2025), ACP Blair & Faro (2025), Agora Marro et al. (2024) SLIM AGNTCY (2025) provide message exchange and discovery mechanisms, yet they offer no structured approach for sharing the reasoning flexibility that underlies agent decisions.

The fundamental challenge is that LLM agents naturally express decision sensitivity through textual reasoning—assessments like "demand spike seems temporary" or "upstream capacity more constraining than downstream orders"—but no coordination protocol exists for systematically sharing and aggregating such qualitative sensitivities. When agents share only final decisions without the reasoning flexibility behind them, coordination becomes brittle: locally rational choices lead to collective failures as agents lack visibility into others' reasoning processes and constraints Stone & Veloso (2000); Lee et al. (1997); Cooper et al. (1990).

We introduce the Ripple Effect Protocol (REP), a coordination protocol specifically designed for LLM-based agents where agents share not only their decisions but also lightweight textual sensitivities—natural language signals that express how their choices would change if key environment

variables shifted. These sensitivities ripple through local networks via existing transport protocols like SLIM, enabling groups to align faster and more stably than with decision-only communication.

REP's design separates agent cognition from protocol coordination. Agents remain responsible for local reasoning and policy evaluation using their native LLM capabilities, while the protocol handles sensitivity aggregation and consensus mechanisms. This separation ensures that REP can coordinate heterogeneous agents—from different LLM architectures to hybrid rule-based systems—without constraining their internal decision-making processes.

The key insight underlying REP is that scalable coordination for LLM agents requires protocol-mediated sharing of decision flexibility rather than structured interaction frameworks. REP provides this coordination intelligence as a protocol primitive, enabling diverse agents to align their actions at scale without constraining their natural language reasoning capabilities.

We demonstrate REP across three coordination domains that represent fundamental challenges in distributed agent systems: supply chain information cascades (Beer Game) Sterman (1989), preference aggregation in sparse social networks (movie coordination), and sustainable resource allocation under competing incentives (Fishbanks) Sterman & King (2017). Across these experiments, REP demonstrates substantial improvements in coordination accuracy compared to traditional agent-to-agent communication, while showing how textual sensitivity sharing accommodates the natural language reasoning of LLM-based agents.

## 2 BACKGROUND AND RELATED WORK

### 2.1 PROTOCOL STACK FOR LLM AGENTS

Recent work has produced a set of communication protocols that enable LLM agents to interoperate across platforms. The Model Context Protocol (MCP) standardizes how agents expose tools and share resources. Cross-agent messaging frameworks such as Google's Agent-to-Agent (A2A) a2aproject (2025), IBM's ACP, the Agent Network Protocol (ANP) Ehtesham et al. (2025), and research systems like Agora Marro et al. (2024) support discovery, authenticated capability exchange, and task hand-offs. For performance-critical settings, SLIM (Secure Low-Latency Interactive Messaging) provides a gRPC-based transport with multicast and publish-subscribe patterns AGNTCY (2025). These protocols form the messaging layer of the emerging Agentic Web, analogous to TCP/IP for networking. They ensure reliable message delivery, routing, and security, but stop short of coordination. While agents can exchange decisions or reasoning traces, there is no mechanism to aggregate information or ensure collective convergence. This creates a gap between communication and true multi-agent coordination.

### 2.2 COORDINATION IN MULTI-AGENT LLM SYSTEMS

Prior work on multi-agent LLMs has focused on centralized orchestration, where frameworks such as AutoGen Wu et al. (2023), MetaGPT Hong et al. (2023), CrewAI Moura, and Swarm OpenAI (2024) enforce structure through shared prompts, memory, and control. These systems provide role definitions, task hand-offs, structured workflows, and stopping criteria. These assumptions work well within a single organization but do not generalize to open networks where agents are independently owned.

For decentralized settings, teams often reuse the same communication protocols introduced above (e.g., A2A, ACP, ANP) and rely entirely on the LLMs themselves to negotiate through free-form reasoning and message exchange. However, without explicit coordination mechanisms, this approach is fragile: agents fall into infinite loops, oscillating handoffs, or fail to detect completion Zhang et al. (2024); Louck et al. (2025); Singh (2025). For example, two agents negotiating a task may repeatedly hand it back and forth, each waiting for confirmation from the other, creating infinite loops that consume resources without ever reaching completion. These issues worsen as the number of agents grows, highlighting that message exchange alone cannot ensure progress or stability.

Classical distributed mechanisms—such as consensus protocols Ongaro & Ousterhout (2014); Castro & Liskov (1999), voting rules Arrow (1951); Sen (1970), and auction mechanisms Cramton et al. (2006) provide such structure in traditional systems but assume static preferences and shared

objectives. They cannot capture the dynamic, textual, and partially aligned reasoning of LLM agents operating in domains with diverse incentives. Recent studies on LLM voting and census-level preference aggregation Yang et al. (2024); Chopra et al. (2025) explore some of these mechanisms in controlled, centralized settings, but real-time decentralized coordination remains an open challenge.

This gap motivates the need for protocols that operate above message exchange to provide true coordination capabilities. REP addresses this challenge by introducing a dedicated coordination layer for the Agentic Web, enabling agents to move from merely communicating to collectively aligning their actions in open, decentralized networks.

### 2.3 SENSITIVITY SHARING IN DISTRIBUTED SYSTEMS

Sensitivity sharing has long been used to enable coordination without central control. In distributed optimization, nodes exchange gradients or dual variables rather than final solutions, as in ADMM and gradient tracking algorithms Boyd et al.; Nedic et al. (2014). In decentralized simulations, agents similarly exchange lightweight signals that summarize local state instead of sharing full datasets. This enables global calibration while preserving privacy—for example, contact tracing protocols that share infection-state updates without revealing individual health data Chopra et al. (2024); Attrapadung et al. (2024).

These approaches assume numerical signals and shared global objectives. In contrast, LLM-based agents reason in natural language, are independently developed, and often operate with partially aligned or competing incentives. Existing sensitivity sharing techniques cannot capture this qualitative reasoning or scale to open, heterogeneous networks. REP builds on this lineage by leveraging textual sensitivity Yuksekgonul et al. (2025) as a coordination primitive, providing the missing layer above current messaging protocols and enabling large populations of LLM agents to coordinate without centralized control.

## 3 RIPPLE EFFECT PROTOCOL

REP introduces a new coordination layer for multi-agent systems, enabling agents to move beyond merely exchanging messages to aligning their actions in open, decentralized networks. At its core, REP defines a set of shared *coordination variables* and a process for *sensitivity sharing*, where agents communicate lightweight signals indicating how their decisions would change under different conditions. These sensitivities are aggregated across local neighborhoods using either gradient-style updates or LLM-based synthesis, producing a compact coordination state that guides future decisions. This structure allows diverse LLM agents to coordinate without centralized control, providing scalability and stability missing from current systems.

REP cleanly decouples internal agent cognition from the external coordination mechanism:

- **Agents think:** Each agent uses its internal reasoning (LLM or rule-based) to decide on domain actions and generate sensitivities—signals describing how its decision would change under different environmental conditions.
- **Protocols coordinate:** REP manages the exchange and aggregation of these sensitivities, updating a shared set of *coordination variables* that influence future decisions.

### 3.1 WORKFLOW OVERVIEW

REP operates over a network $G = (V, E)$, where each node $i \in V$ represents an agent and edges $E$ indicate direct communication links between neighbors $N_i = \{j \in V \mid (i, j) \in E\}$. Time proceeds in discrete **rounds** $t = 1, 2, \ldots$, with each round consisting of four steps:

1. **Receive Messages:** Each agent $i$ collects messages $M_i^t = \{(d_j^t, s_j^t) : j \in N_i\}$ containing neighbor decisions $d_j^t$ and sensitivities $s_j^t$.

2. **Generate Decision & Sensitivity:** The agent applies its policy $\pi_i$ to the current coordination variables $\theta_i^t$ and private constraints $c_i$ to produce:

$$(d_i^t, s_i^t) = \pi_i(\theta_i^t, c_i)$$

where $d_i^t$ is the domain action (e.g., order quantity, vote, fleet deployment) and $s_i^t$ is a signal describing how the decision would change if conditions shifted.

3. **Aggregate Neighbor Sensitivities:** REP combines neighboring sensitivities to update local coordination variables:

$$\theta_i^{t+1} = \theta_i^t + \text{Agg}_{j \in N_i}(s_j^t)$$

Sensitivities may be numeric or textual, with REP applying either standard optimization rules or LLM-based synthesis as needed.

4. **Consensus (Optional):** In domains requiring group agreement, REP applies deterministic rules such as coordinate-wise median to converge on shared proposals.

**Example:** In a supply chain, an agent's decision might be to order 120 units. Its sensitivity could read: *"If demand increases by 10%, increase order by 15 units; if upstream capacity improves, decrease by 5 units."* These sensitivities are aggregated by neighbors, producing updated coordination variables like `TARGET INVENTORY` and `ORDER ADJUSTMENT FACTOR`, which then shape the next round's decisions.

## 3.2 COORDINATION AND AGGREGATION

At each round $t$, every agent $i$ maintains a local coordination state $\theta_i^t$ that influences its next-round decision. Using its internal policy $\pi_i(\theta_i^t, c_i)$, the agent produces two outputs: a domain action $d_i^t$ and a *sensitivity message* $s_i^t$ describing how $d_i^t$ would change under counterfactual conditions. These sensitivities are exchanged among neighbors $\mathcal{N}_i$ and combined to update the agent's local state.

REP frames this process as a generalized gradient step:

$$\theta_i^{t+1} = \theta_i^t - \eta \cdot g_i^t,$$

where $\eta$ is a step size and $g_i^t$ is a "gradient-like" signal representing the aggregated influence of neighboring agents:

$$g_i^t = \text{Agg}_\phi\big(\{s_j^t : j \in \mathcal{N}_i\}, \theta_i^t\big).$$

The operator $\text{Agg}_\phi$ is parameterized by a synthesis function $\phi$, which determines how sensitivities are integrated. REP is modality-agnostic: in some domains, $\phi$ is a numeric rule such as weighted averaging or distributed gradient descent; in others, $\phi$ is a language model that synthesizes free-form reasoning into structured, low-dimensional updates. This approach is related to recent methods such as TextGrad Yuksekgonul et al. (2025), which treat natural language as a medium for computing gradient-like updates, but in REP these signals are used to coordinate *populations of agents* rather than optimize a single model.

This unified formulation allows REP to seamlessly handle both quantitative and qualitative coordination signals without changing the overall update rule. For example, $s_j^t$ might encode either production adjustments in a supply chain or price/time flexibility in group scheduling.

In many domains, these local updates alone are sufficient. However, some tasks require the entire population to agree on a shared proposal, such as selecting a common meeting time or price. In these cases, REP adds a second, global consensus step after local aggregation:

$$\bar{\theta}^{t+1} = \text{Median}\big(\{\theta_i^{t+1} : i \in \mathcal{V}\}\big),$$

where $\mathcal{V}$ is the set of all agents. This coordinate-wise median prevents extreme local outliers from destabilizing the collective decision while retaining diversity of preferences. The resulting two-level process enables REP to span a broad spectrum of coordination problems, from local alignment in sequential supply chains and resource management to explicit consensus formation in distributed decision-making.

## 3.3 IMPLEMENTATION ARCHITECTURE

REP is implemented as a lightweight coordination layer that wraps existing agents without modifying their internal policies. A `REPClient` manages message exchange, sensitivity aggregation, and updates to coordination variables. The design is intentionally modular: the transport backend, aggregation rule, and consensus mechanism can each be configured independently. This makes REP

portable across domains and independent of any single communication protocol or decision rule. For example, we use SLIM for multicast messaging and coordinate-wise median for consensus, but other systems or rules can be substituted without changing the protocol logic.

**Configuration Interface.** Each agent is initialized by specifying which implementations to use for these three components:

```
rep_client = rep.configure(
    agent=llm_agent,
    transport="slim",               # any messaging backend
    updater="textual_grad",         # or "numerical_grad"
    consensus="median_coordinate"   # or other rules
)
```

**Protocol Execution.** During each round, agents run the same loop in parallel: receiving messages, generating local decisions and sensitivities, broadcasting them to neighbors, and performing local and (optionally) global aggregation:

```
# Protocol execution
neighbor_msg, neighbor_sensitivity = rep_client.receive()
rep_client.sync(neighbor_sensitivity)  # aggregate neighbor sensitivities
decision, sensitivity = rep_client.decide()
rep_client.send(decision=decision, sensitivity=sensitivity)
```

When textual aggregation is enabled, domain-specific prompt templates guide the LLM in synthesizing free-form reasoning into structured updates. This allows the same core protocol to support diverse coordination problems—from supply chain optimization to distributed consensus—without changing agent logic or protocol implementation. We provide details about prompt specification in appendix A.

## 4 EXPERIMENTAL SETUP

We evaluate REP across three domains that systematically span different network structures, incentive patterns, and coordination challenges. The Beer Game models sequential coordination in linear supply chains (4–8 agents) with aligned incentives but delayed information flows that create bullwhip effects. Movie Coordination captures consensus formation in sparse social networks (5–20 agents), where heterogeneous preferences must be reconciled despite limited connectivity. Fishbanks examines resource allocation in fully connected networks (5–8 agents) with directly competing incentives, where individual profit maximization conflicts with collective sustainability. Together, these environments test REP across a spectrum of coordination conditions: information cascades versus multi-hop influence versus broadcast interactions; aligned, heterogeneous, and conflicting incentives; and network topologies ranging from linear to sparse to dense.

Table 1 summarizes the core elements of each domain, including the decisions made by agents, the coordination variables introduced by REP, and the structural properties of each environment.

Table 1: Agent decision variables, coordination variables, and challenges for each domain.

| Domain | Agent Decisions | Coord. Variables | Network | Challenge |
|---|---|---|---|---|
| **Beer Game** | Order quantities per round | Target inventory levels | Linear chain | Information cascades and bullwhip effect |
| **Fishbank** | Fleet deployment across regions | Sustainable quotas | Fully connected | Mixed-motive allocation and over-exploitation |
| **Movie Scheduling** | Participation (Y/N) or local preference votes | Preferred price/time thresholds | Sparse small-world network | Consensus formation with heterogeneous preferences |

Our baseline represents the current state of practice in open LLM-agent networks. We use an Agent-to-Agent (A2A) approach where agents share their final decisions and free-form reasoning traces. This mirrors the message exchange patterns supported by protocols such as A2A and ACP, and the coordination style seen in multi-agent orchestration frameworks like MetaGPT, CrewAI, and OpenAI Swarm today. For example, in the Beer Game, an upstream agent might message, "Ordering 12 units because inventory is low." While this communication enables neighbors to understand intent, it cannot be systematically aggregated to update shared coordination variables. Both REP and the baseline use identical transport, consensus rules, LLM model, and network topologies, ensuring that performance differences are solely due to REP's introduction of structured sensitivity sharing.

All experiments use Claude-3-Haiku-20240307 (temperature = 0.1, max_tokens = 300) to ensure consistent reasoning power across conditions. Each simulation runs for 20 rounds, with 3–5 trials per configuration to account for stochasticity. Coordination accuracy is reported using domain-specific metrics, detailed in the sections describing each experiment. Final results are aggregated across trials with mean and standard deviation reporting. To validate communication efficiency, we scale the number of agents from 10 to 200 and measure wall-clock coordination time. This test demonstrates that REP's structured design, combined with SLIM's low-latency multicast messaging, supports efficient coordination for population scale.

## 5 SUPPLY CHAIN COORDINATION (BEER GAME)

The Beer Game Sterman (1989) is a canonical benchmark for studying coordination failures in sequential supply chains. It consists of four stages—retailer, wholesaler, distributor, and manufacturer—arranged in a linear chain. In each round, every agent observes only its local inventory and orders from its immediate neighbor, deciding how many units to order upstream. Fixed lead times delay feedback, meaning agents cannot immediately observe the downstream consequences of their actions. This limited visibility causes small fluctuations in customer demand to amplify as they propagate upstream, producing the well-known bullwhip effect Lee et al. (1997).

In the baseline Agent-to-Agent (A2A) approach, agents share both order quantities and free-form reasoning traces describing the rationale behind those decisions. While this helps neighbors understand why an action was taken, it does not capture how the agent's behavior would change under different future conditions. For instance, when a retailer places an unusually large order and explains it as "inventory low due to a local demand spike," upstream agents still lack visibility into whether this action represents a temporary fluctuation or a longer-term shift. Without explicit modeling of decision flexibility, these free-form messages cannot be aggregated into a stable shared understanding, leaving the system vulnerable to information cascades and runaway amplification.

REP introduces structured sensitivity messages, where each agent communicates how its decision would adapt under key environmental changes. For example, "If demand increases by 10%, increase order by 15 units. If upstream capacity improves, reduce order by 5 units. Current spike seems temporary." These sensitivities are combined across neighboring agents into low-dimensional coordination variables, such as TARGET_INVENTORY and ORDER_ADJUSTMENT_FACTOR, which guide subsequent decisions. By circulating structured decision flexibility rather than only final actions and free-form text, agents gain predictive context that allows them to distinguish transient noise from structural change, reducing instability.

**Insight 1: REP mitigates bullwhip effects.** Under identical LLM agents and network structures, REP reduces total supply chain cost by 41.8%, from \$7,300 (A2A) to \$4,251, and stabilizes demand shocks within 3–4 rounds, compared to 10+ rounds under A2A (Fig. 1(a-b)). REP's structured signals prevent the repeated over- and under-reactions that drive oscillations in traditional systems, leading to rapid convergence after initial disturbances.

**Insight 2: Textual sensitivities outperform numerical gradients** REP supports both numerical aggregation—where sensitivities are treated as structured derivatives—and textual aggregation, where natural language reasoning is synthesized into compact updates. Textual sensitivities yield lower total cost ($4,251$ vs $4,680$, a 9.2% improvement; Fig. 1(c) by capturing nuanced causal relationships, such as supplier behavior or distinguishing short-term volatility from systemic shifts, that numerical signals cannot express.

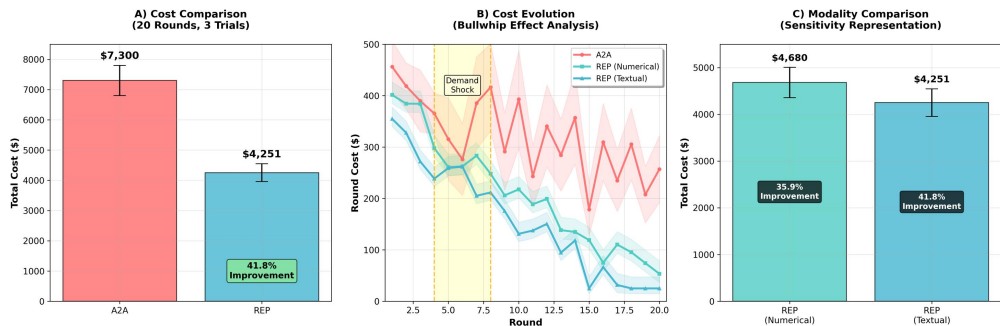

Figure 1: Supply chain coordination outcomes over 20 rounds with a demand shock introduced at round 4, where customer demand doubles. (A) Baseline A2A shows persistent bullwhip oscillations as agents repeatedly over- and under-react to incomplete information. REP reduces total supply chain cost by 41.8%, from $7,300 (A2A) to $4,251 (B) REP rapidly stabilizes within 3–4 rounds, preventing runaway amplification. (C) REP textual aggregation outperforms numerical aggregation, achieving lower total costs by capturing richer causal reasoning.

# 6    RESOURCE ALLOCATION (FISHBANKS)

Resource allocation under sustainability constraints exemplifies the tragedy of the commons, where individually rational decisions lead to collectively irrational outcomes Hardin (1968); Ostrom (1990). The Fishbanks simulation models this dynamic through 12 fishing companies operating in shared waters Sterman & King (2017). Each company aims to maximize profits by deploying its fleet to different fishing zones, but uncoordinated strategies lead to overfishing and eventual resource collapse, harming all participants.

In the baseline Agent-to-Agent (A2A) approach, companies share their deployment decisions and reasoning—such as the number of boats sent to each zone—based on local observations of catch rates and market prices. Without additional structure, agents cannot determine whether others are pursuing short-term opportunistic gains or making long-term sustainability commitments. This information isolation drives competitive escalation: each company acts defensively, deploying aggressively to avoid being left behind, which accelerates resource depletion.

REP extends this setting by allowing companies to communicate textual sensitivities about resource conditions and coordination willingness alongside deployment decisions. For example, "Fish population shows recovery signs based on catch composition. Fleet maintenance costs justify a conservative approach if others also coordinate.". These sensitivities capture how decisions would shift under different environmental or strategic conditions, enabling agents to infer whether others are willing to restrain harvesting for collective benefit. Aggregating these signals across companies provides a shared picture of ecosystem health and alignment, supporting voluntary, decentralized coordination without a central controller.

**Insight 3: Coordination under competing incentives.**    Across 8-season experiments with 12 LLM agents, REP achieves 25.2% sustainability improvement and 28.9% better population health while preventing the financial losses experienced by A2A agents (-2.5%). This demonstrates REP's capability to coordinate in tragedy-of-commons scenarios—the most challenging multi-agent setting where individual and collective objectives directly conflict. Unlike traditional approaches that result in resource depletion harming all participants, REP enables win-win outcomes through voluntary coordination. This is visualized in figure 2

**Insight 4: Transparency enables conditional cooperation.**    The shared sensitivities include both ecosystem assessments and signals of willingness to cooperate. Companies can condition their behavior on others' stated intentions—e.g., "We can maintain a conservative fleet size if others do the same". This transparency allows trust to emerge without explicit enforcement, shifting agents from

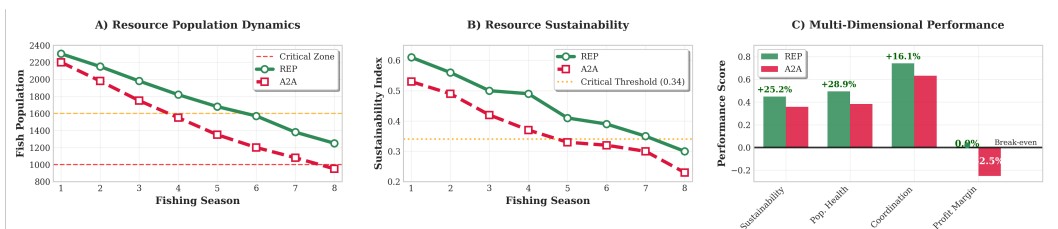

Figure 2: Fishbanks simulation with 12 LLM agents over 8 seasons. A demand for fish remains constant while overharvesting leads to eventual depletion. (A) Fish population dynamics showing delayed resource collapse under REP compared to rapid decline in the A2A baseline. (B) Sustainability index with a critical viability threshold at 0.35; REP maintains the resource above this threshold 2 seasons longer than A2A. (C) Comparison of final outcomes across normalized metrics and profit margins. REP improves sustainability (+25.2%), population health (+28.9%), and coordination (+16.1%) while avoiding the financial losses (-2.5%) experienced by A2A agents. Results averaged over five trials.

pure profit-maximization toward coordinated stewardship of the shared resource (detailed sensitivity examples in Appendix B).

## 7    PREFERENCE AGGREGATION (MOVIE COORDINATION)

Group decision-making in sparse social networks presents a fundamental coordination challenge: agents must converge on a shared outcome despite heterogeneous preferences and limited connectivity. Each agent represents an individual with distinct budget and scheduling constraints, communicating only with immediate neighbors. As a result, preferences must propagate indirectly through the network.

In the baseline Agent-to-Agent (A2A) setting, agents exchange their final participation choices and free-form reasoning. Without structured aggregation, this leads to information isolation: small clusters of neighbors may agree locally, but there is no mechanism to align the full network. This produces fragmented, unstable outcomes, where local proposals conflict and the group fails to converge on a global consensus.

REP introduces a structured preference-sharing process that unfolds in two stages during each round. In the first stage, every agent communicates both its current participation decision and a set of preference sensitivities that capture how its utility would change under adjustments to the shared coordination variables. These sensitivities can take numerical form, such as $\partial U/\partial \text{price} = -0.8$ to indicate high price sensitivity, or textual form, such as *"Price matters more than timing — strongly budget constrained."* Messages from neighboring agents are then aggregated for the two global coordination variables, **TIME** and **PRICE**, with local updates nudging these variables toward values that reflect the collective preference landscape. In the second stage, agents vote again on the updated proposal, and REP applies a median consensus rule to produce the next shared state. This consensus mechanism provides stability by preventing extreme local outliers from destabilizing the group decision. Through repeated rounds, sensitivities ripple across multiple network hops, allowing the population to converge even when many agents are not directly connected.

**Insight 5: Coordination effectiveness across network sparsity.** We evaluate REP on a 20-agent network under three connectivity regimes: fully connected, 30% sparse, and 60% sparse. **REP** achieves **70–75% consensus** across all conditions (Fig. 3(left)), with only gradual increases in convergence time: Round 4 → Round 6 → Round 9 as connectivity decreases. This resilience arises because sensitivity signals propagate indirectly, allowing agents to align even when some connections are missing. In contrast, **A2A** fails to reach meaningful agreement, plateauing at **35% maximum consensus** even in fully connected networks. Without structured aggregation, each agent is overwhelmed by conflicting inputs, leading to cognitive overload and preventing the system from integrating distributed reasoning into a stable global outcome.

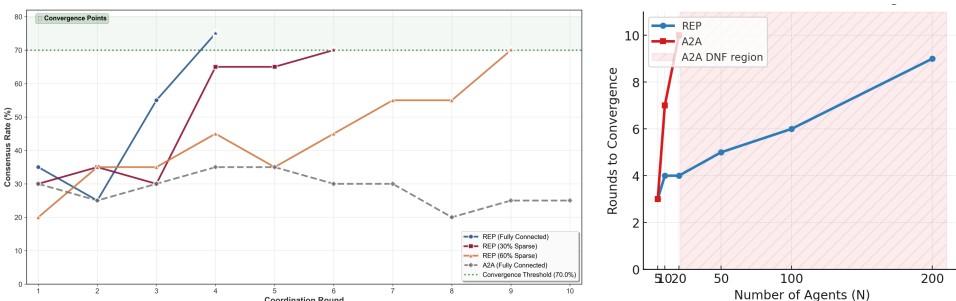

Figure 3: REP performance under network sparsity and population scaling. (Left) REP maintains effective coordination even in sparse social networks (30% and 60% sparse), with only gradual increases in convergence rounds (4 → 6 → 9) while sustaining 70–75% consensus. Stars mark convergence points where the 70% threshold is reached. In contrast, A2A fails to achieve meaningful coordination even when fully connected, showing that sensitivity sharing is essential for global alignment. (Right) As population size grows from 5 to 200 agents, REP continues to converge reliably (3–15 rounds), while A2A requires 7–10 rounds at small sizes and fails entirely beyond 20 agents (shaded DNF region). Together, these results demonstrate REP's robustness to both sparse connectivity and large-scale populations.

**Insight 6: Scalability to larger populations** To evaluate REP's scalability, we increase the network size from 5 to 200 agents at fixed sparsity. Fig. 3(right) projects the number of rounds required to reach convergence as population size grows. REP maintains stable performance: convergence occurs in just 3–9 rounds across the entire range, with only modest growth as the network scales. In contrast, A2A requires 7–10 rounds even for small populations ($\leq$20 agents) and fails to converge entirely beyond this point, as indicated by the shaded DNF region. Because per-round message cost grows near-linearly with N under SLIM multicast, fewer rounds directly translate to lower total communication. Wall-clock profiling further shows that communication is negligible: at 200 agents, sensitivity sharing is 3% of runtime, with the rest dominated by LLM inference (38%) and wait time (59%). Thus, REP's scalability is promising for designing an agentic web.

# 8 CONCLUSION AND FUTURE WORK

We introduced the Ripple Effect Protocol (REP), a lightweight coordination layer that enables LLM-based agents to share sensitivities rather than only final decisions. By operating above existing messaging frameworks like A2A and SLIM, REP turns coordination into a protocol primitive, allowing diverse agents to align their actions across different network structures, incentive regimes, and temporal dynamics. Across three domains—supply chains, shared-resource management, and group decision-making—REP consistently outperformed traditional decision-only communication by accelerating convergence, improving stability, and enabling coordination at population scale.

Our current work assumes cooperative, non-malicious agents and synchronous interactions, which simplifies reasoning but limits applicability to fully open, decentralized networks. Future work will extend REP along two key directions. First, we plan to develop Byzantine fault-tolerant mechanisms that allow coordination even when some agents misreport sensitivities or behave adversarially, ensuring robustness in competitive or adversarial settings. Second, we will generalize REP to support asynchronous multi-step interactions, where agents operate on partially overlapping timelines and must integrate information without requiring strict round synchronization. These enhancements will broaden REP's applicability to real-world, large-scale deployments such as financial markets, distributed infrastructure, and cross-organization collaboration. We will release the REP SDK along with this paper and provide a sandbox environment to test scalability of coordination and transport protocols. We hope our research accelerates the development of the Agentic Web.

ETHICS STATEMENT

This research adheres to the ICLR Code of Ethics and follows all principles of responsible AI research. Our work contributes to scientific excellence by providing rigorous empirical evaluation and transparent methodology. The research poses no direct societal harm and aims to improve reasoning capabilities in language models for beneficial applications. We acknowledge potential dual-use considerations of improved reasoning systems and encourage responsible deployment. All experiments were conducted on publicly available models and benchmarks with proper attribution. No human subjects were involved in this research.

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

## A APPENDIX A: SYSTEM PROMPT IMPLEMENTATION DETAILS

This appendix describes the prompt structure used in our experiments, highlighting how REP differs from the A2A baseline. Both protocols follow a two-API-call process each round:

1. **API Call 1**: Each agent outputs its decision. REP also includes a *sensitivity message* describing how its decision would change under different conditions.

2. **API Call 2**: Each agent updates internal coordination variables by aggregating neighbor information. REP uses neighbors' reasoning, while A2A relies only on numerical decisions.

### A.1 TWO-API-CALL ARCHITECTURE

Figure 4 shows the workflow for the Fishbank domain, contrasting REP and A2A information flow.

```
REPL Protocol - API Call 1                              A2A Baseline - API Call 1
Decision + Sensitivity                                  Decision + Reasoning

You are the AI decision-maker for GlobalMega_Corp,      You are the AI decision-maker for GlobalMega_Corp,
a Deep-sea industrial fishing company in a 12-company   a Deep-sea industrial fishing company in a 12-company
shared fishbank environment.                            shared fishbank environment.

PROTOCOL UNDERSTANDING (Local Variables):               NUMERICAL NEIGHBOR PATTERNS (A2A Information):
- Sustainable quota estimate: 180.0 fish                - Number of other companies: 11
- Population health indicator: 0.85/1.0                 - Average neighbor boats deployed: 4.2
- Industry coordination confidence: 0.4/1.0             - Neighbor deployment variance: 1.8
- Coalition formation potential: 0.6/1.0                - Total neighbor boats: 46

NEIGHBOR INTELLIGENCE (Reasoning):                      LOCAL PERFORMANCE INDICATORS:
Company 1: 4 boats - "Considering sustainability        - Market trend indicator: 0.1 (-1=declining, 1=growing)
concerns and market volatility, reducing deployment     - Competition pressure: 0.7 (0=low, 1=high)
to preserve long-term viability..."                     - Personal success rate: 0.6
Company 2: 6 boats - "High coordination confidence      - Resource abundance estimate: 0.5
suggests collective restraint strategy is working..."
                                                        12-COMPANY A2A CONSIDERATIONS:
12-AGENT COORDINATION CONSIDERATIONS:                   Based on numerical patterns only (no textual reasoning from other
1. Navigate complexity of 12-company dynamics and       companies):
coalitions                                              1. Analyze the numerical deployment patterns of the 11 other companies
2. Identify alliance patterns among companies           2. Consider your company's risk tolerance and sustainability values
3. Balance individual optimization with collective      3. Optimize based on market conditions and operational costs
sustainability
                                                        TASK:
TASK:                                                   Decide how many boats to deploy this season (1-10) based solely
1. Decide how many boats to deploy this season (1-10)   on numerical information and your company profile.
2. Provide detailed sensitivity analysis considering
12-company dynamics                                     Your response format:
                                                        DECISION: [number]
Your response format:                                   REASONING: [brief explanation of your numerical analysis and
DECISION: [number]                                      decision logic]
SENSITIVITY: [detailed reasoning about your decision,
coordination analysis, sustainability assessment,
coalition considerations]
```

```
REPL Protocol - API Call 2                              A2A Baseline - API Call 2
Sensitivity Aggregation                                 Coordination Update

SENSITIVITY AGGREGATION FOR LOCAL VARIABLE UPDATE:      I am GlobalMega_Corp coordinating with my neighbors.

YOUR SENSITIVITY: "Considering sustainability concerns  MY CURRENT DEPLOYMENT STRATEGY:
and market volatility, reducing deployment to preserve  - Current boats deployed: 5
long-term viability for our 12-company ecosystem"       - Current market assessment: Moderate competition
                                                        - Current sustainability stance: Profit-focused
NEIGHBOR SENSITIVITIES FROM CONNECTED AGENTS:
Neighbor 1: "Considering sustainability concerns and    NEIGHBORS' STRATEGIES AND DECISIONS:
market volatility..."                                   Neighbor 1: 4 boats - "Conservative approach"
Neighbor 2: "High coordination confidence suggests      Neighbor 2: 6 boats - "Aggressive expansion"
collective restraint..."                                Neighbor 3: 3 boats - "Sustainability focused"
Neighbor 3: "Resource abundance estimates declining,    Neighbor 4: 7 boats - "Market opportunistic"
need strategic coordination..."                         [...neighbor deployment decisions and reasoning...]
Neighbor 4: "Market conditions favor conservative
deployment approach..."                                 ENVIRONMENT FEEDBACK:
[...connected neighbors' sensitivity reasoning...]      - Market price: $12.50/fish
                                                        - Industry total catch: 52 fish
ENVIRONMENT OUTCOME:                                    - Competition level: High
- Fish population change: -15%
- Total industry catch: 52 fish                         TASK: Based on neighbor decisions and market conditions, adjust
- Market price: $12.50/fish                             my deployment strategy. What should my new deployment level be?
- Sustainability index: 0.73
                                                        FORMAT:
TASK: Based on neighbor sensitivities and environmental STRATEGY: [deployment adjustment reasoning]
outcomes, update your local variables:                  BOATS: [new deployment number]
1. sustainable_quota_estimate: Your personal estimate
   of sustainable catch limit
2. population_health_indicator: Your assessment of fish
   stock health (0-1)
3. coordination_confidence: Your trust in neighbor coordination (0-1)
4. resource_stress_estimate: Your local assessment of resource
   pressure (0-1)

FORMAT: Provide updated local variable values as JSON.
```

Figure 4: Two-API-call structure for the Fishbank experiment. API Call 1 produces agent decisions, and in REP also generates sensitivity reasoning. API Call 2 aggregates neighbor information to update local variables.

### A.2 PROTOCOL VARIABLES

Each REP agent maintains a vector of coordination variables:

$$\theta = \{\text{QUOTA\_ESTIMATE, POPULATION\_HEALTH, COORDINATION\_CONFIDENCE, RESOURCE\_STRESS}\},$$

representing the agent's understanding of the shared resource. These are updated through neighbor reasoning in REP, whereas A2A has no equivalent global state.

Table 2: Comparison of REP and the A2A baseline. Both use identical network, transport, model, and consensus settings; the only difference is REP's structured sensitivity sharing and aggregation.

| Aspect | REP | A2A Baseline |
|---|---|---|
| Transport Layer | SLIM multicast messaging | SLIM multicast messaging *(same as REP)* |
| Consensus Rule | Coordinate-wise median | Coordinate-wise median *(same as REP)* |
| LLM Hyperparameters | Claude-3-Haiku, temp = 0.1, max tokens = 300 | Same model and hyperparameters |
| Information Sharing | Decisions + structured sensitivities (decision flexibility) | Final decisions + free-form reasoning text only |
| Learning Dynamics | Aggregated neighbor sensitivities for collective updates | Local reasoning only, no structured aggregation |
| Coordination Variables | $\theta = \{quota, health, confidence, stress\}$ maintained globally | None beyond private local state |
| Decision Process | `DECISION + SENSITIVITY` | `DECISION + REASONING` |

## A.3 REP VS A2A COMPARISON

## A.4 REPRESENTATIVE PROMPTS (FISHBANK EXAMPLE)

### A.4.1 REP: API CALL 1 (DECISION + SENSITIVITY)

**Input:** Neighbor decisions and reasoning about sustainability, market conditions, and coordination behavior.
**Task:** Decide how many boats to deploy and describe what changes would make you increase or decrease this number.
**Output Format:**

```
DECISION: [integer number of boats]
SENSITIVITY: [detailed explanation of triggers and thresholds]
```

### A.4.2 REP: API CALL 2 (SENSITIVITY AGGREGATION)

**Input:** Neighbor sensitivities describing ecosystem conditions and coordination signals.
**Task:** Update your local variables:

- quota_estimate
- population_health (0–1)
- coordination_confidence (0–1)
- resource_stress (0–1)

**Output:** JSON with updated values for each variable.

### A.4.3 A2A: API CALL 1 (DECISION ONLY)

**Input:** Numerical summaries only (e.g., total boats, averages, variance).
**Output Format:**

```
DECISION: [integer number of boats]
REASONING: [brief numerical explanation]
```

Full prompt templates and model hyperparameters will be included in the open-source release.

## B APPENDIX B: TEXTUAL SENSITIVITY EXAMPLES

REP's advantage in tragedy-of-commons domains comes from the **context-rich signals** encoded in textual sensitivities. Rather than only reporting final actions (e.g., number of boats deployed), REP

agents explain *why* they acted, sharing assessments of resource health, willingness to cooperate, and strategic positioning. This context enables other agents to distinguish opportunistic behavior from genuine coordination, breaking the tragedy-of-commons dynamic.

Below are representative outputs from the 12-agent Fishbank simulation, illustrating how sensitivities capture collective reasoning.

### B.1 RESOURCE SUSTAINABILITY ASSESSMENTS

Agents share localized ecological observations to create a distributed "sensor network":

> **EcoSustainable_Fisheries (3 boats):**
> *"Sustainable quota estimate of 180 fish and population health indicator of 0.85 suggest a stable condition. Deploying 3 boats balances steady income with preserving future yields."*

**Impact:** Creates shared understanding of ecological constraints beyond individual observations.

### B.2 COORDINATION WILLINGNESS SIGNALS

Agents reveal when they are willing to sacrifice short-term profits for collective benefit:

> **GlobalMega_Corp (5 boats):**
> *"We will lead coalition efforts to align individual goals with shared sustainability objectives."*

**Impact:** Builds trust through explicit conditional cooperation.

### B.3 STRATEGIC AND COMPETITIVE REASONING

Agents track competitive clusters and potential alliances:

> **Traditional_Family_Fleet (3 boats):**
> *"Fleet sizes differ significantly (GlobalMega_Corp: 5 boats vs EcoSustainable_Fisheries: 3 boats), indicating emerging competitive clusters we can help bridge."*

**Impact:** Surfaces multi-agent complexity, enabling nuanced coordination strategies.

### B.4 LINKING TO MEASURABLE OUTCOMES

These sensitivities translate into measurable benefits:

- Final fish population was **1,235** under REP vs **958** under A2A — a **289-fish preservation gain** or a **25.2% sustainability improvement**.

**Key takeaway:** By understanding each other's reasoning and commitments, agents maintain shared resources rather than depleting them.

