# OpenReview forum: "Ripple Effect Protocol"
_ICLR.cc/2026/Conference — Submitted to ICLR 2026_

### Official Review · Reviewer_kJcU · 2025-10-30

**Soundness:** 2
**Presentation:** 2
**Contribution:** 3
**Rating:** 4
**Confidence:** 4

**Summary:**

The paper proposes Ripple Effect Protocol (REP), a coordination layer above existing agent messaging (e.g., A2A/ACP). Instead of sharing only final decisions, agents also share lightweight “sensitivities” such as how their action would change under counterfactuals. REP aggregates these (numeric or textual) across neighbors and may apply an optional consensus step. Evaluations span Beer Game (supply chain), Fishbanks (commons), and Movie Scheduling (consensus in sparse graphs), showing stability and scaling gains over a decision-only A2A baseline.

**Strengths:**

REP makes a clear protocol-level contribution by decoupling agent cognition from coordination and plugging into existing transports (e.g., SLIM) while permitting either numeric or LLM-based aggregation. This is a good and novel motivation and a significant lacking aspect in existing protocols. The empirical analysis also provide two key insights: 1) Providing insights into where existing protocols (A2A) fails 2) Demonstrate how coordination can help in making protocols more scalable.

**Weaknesses:**

The exposition is unclear. The abstract and introduction blur the line between simply appending a “coordination” field to existing structured messages (e.g., in ACP-like protocols) and introducing a new protocol layer with its own state, broadcast/aggregation primitive, and action-update rule. Although section 3 makes the contributions clear but as written, a reader could reasonably conclude that coordination is just another message type rather than a distinct mechanism. Adding a diagram which shows differentiation from existing protcols/how REP complements existing protocols would be helpful and I also suggest updating the abstract and introduction more clearly, clarifying the architectural level difference from existing protocols early on.

The empirical scope also feels narrow. Beyond the two solid takeaways (failure modes of existing protocols and practical scaling benefits), several broader claims are not well supported and appear tightly coupled to the three chosen scenarios. To demonstrate that the layer itself—not merely richer text or longer messages—drives the gains, add targeted ablations: (i) an A2A control that embeds the same coordination content directly in model responses with matched token budgets but no aggregation/update; (ii) removal of the update step (broadcast-only); (iii) variants that drop or swap the aggregation/consensus component; and (iv) limited stress tests across tasks/topologies. These additions would clarify necessity vs. sufficiency of structured coordination and quantify each component’s contribution.

**Questions:**

1. How would REP's be helpful in general tasks apart from the one's demonstrated in the paper?
2. What are the expected advantages of REP’s aggregation and update mechanism compared to simply letting agents decide independently after receiving others’ coordination messages?

---

### Official Review · Reviewer_6Pwk · 2025-10-30

**Soundness:** 3
**Presentation:** 4
**Contribution:** 4
**Rating:** 8
**Confidence:** 4

**Summary:**

The authors develop the Ripple Effect Protocol a novel way for multiple agents to communicate with each other towards the aim of achieving coordination and consensus. They show empirically across three the domains how their work improves above the baselines algorithms.

**Strengths:**

I found the research to be highly novel and interesting. The questions are clearly motivated, and the approach is innovative.

**Weaknesses:**

This was confusing at this point of the paper: "The fundamental challenge is that LLM agents naturally express decision sensitivity through textual
reasoning—assessments like “demand spike seems temporary” or “upstream capacity more constraining than downstream orders”—but no coordination protocol exists for systematically sharing and aggregating such qualitative sensitivities.”

The authors needs to show robustness across both stronger models (e.g., SOTA models) and open weight models for reproducibility. This is really a requirement for acceptance.

The interpretation of the Fishbanks data does not match the results. Looking at Figure 2, it REP just somewhat reduces the decline but the population is just as certaintly headed towards collapse. This is a far cry from “coordinated stewardship of the shared resource” as discussed in the paper.

It would also be useful to cite related work on these kinds of common pool resource problems with multi-agent LLMs e.g., Piatti, Giorgio, et al. "Cooperate or collapse: Emergence of sustainable cooperation in a society of llm agents." Advances in Neural Information Processing Systems 37 (2024): 111715-111759.

**Questions:**

Give some descriptive analysis. What kinds of sensitivities do models tend to use? What sensitiveness are most useful and which are filler? How many are needed to achieve robustness.

How does REP work in open-weight models or SOTA models?

How does it affect scalability. How much overhead does it ad?

---

### Official Review · Reviewer_iupu · 2025-10-31

**Soundness:** 2
**Presentation:** 3
**Contribution:** 2
**Rating:** 4
**Confidence:** 4

**Summary:**

The paper proposes a protocol to share, in an agentic-LLMs network, signals that express how an agent’s actions would vary under uncertainty.

The idea is that an agent, before replying to a message that contains a numerical/discrete value on which they base their response, they compute a “what if” scenario where the value changes and report that in their message.

Experiments are conducted on 3 domains, showing improved performance and communication efficiency.

**Strengths:**

The paper proposes an intuitive solution to an important problem in multi-agent LLMs, i.e., that of propagating the uncertainty an agent has on the exogenous variables of the system.
That is an excellent starting point for a solid paper.
The article is also easy to read, and the technique works well in the 3 domains where it is tested.

**Weaknesses:**

I will call uncertainty the fact that an agent expresses what they would do if the exogenous conditions/variable (the dynamics of the network) change.

My main concern about the paper is that the idea, framework, and implementation, while interesting, are implemented with simplified assumptions about the uncertainty and its estimation; that makes the resulting setting incremental and not realistic (on top of the setting being cooperative, non-malicious and synchronous, which I discuss later).

Two major concerns follow from the previous observation: (1) there is no mechanism or discussion about the computational overhead this system introduces, and (2) to compute the real uncertainty of a model (i.e., something we can really rely on), one has to solve an NP-hard problem.
As regards (1), consider the realistic, yet straightforward case where an agent has to express the uncertainty of its output (that, say, consists of an integer) w.r.t. k exogenous variables that can assume j different values. The model has to compute and communicate the uncertainty for (k*j) different values of the exogenous variable!
In other words, unless the model’s output is discrete and monotonic w.r.t. the dynamics of the network (a *very strong* assumption), computing the uncertainty becomes easily intractable for LLMs, the context length grows very quickly, and the communication overhead dwarfs that of the baseline.

I understand that this computation is not done in practice but “guessed and reported” by an agent, and that is a problem as we cannot rely on that estimation as the real action the model would commit to in case the exogenous conditions change.

Further, suppose one wants to estimate the *real* uncertainty of a model w.r.t. some exogenous parameters. That should be done by either querying the model for many variations of the exogenous variables or, equivalently, computing the Lipschitz constant of the model w.r.t. the exogenous variables, which is intractable and, in most cases (e.g., Transformers), not possible at all.

This, on top of the agents that the authors assumed to be cooperative, non-malicious and synchronous, all characteristics that are non-standard in any multi-agent system.

Last, the technique mimics in spirit but does not cite or mention React. If I look at Section 3, I see a similar framework to reasoning (in your case, about uncertainty and its consequences), then acting (propagating the uncertainty). Consider adding that and the related literature in the revised version of the paper.


Another minor issue is that, if we look at the simplified version of the problem the authors propose, and we assume agents are not malicious and veracious, it seems evident to me that the additional information provides some benefit to the system. The agents have more observability on the state of the other agents, thus they can better coordinate.

To conclude, while I appreciate the effort, I believe the work requires an extension to capture cases where this information is not necessarily veracious, there is no some anti-monotonicity in the agents behaviour (e.g., “agent A will bet 10 if variable X is 1, will bet 15 if it is it is 2, wille bet 0 if it is 3, …”), etc.

**Questions:**

Q1. How would you compute the real uncertainty, in practice, for models like LLMs and under non-monotonic assumptions?

Q2. In complex scenarios where models can lie or just be wrong about their uncertainty estimation (if done via prompting, as in your case), how do we trust that we will have an advantage by propagating potentially wrong information and beliefs?

Q3. How do you think your framework would compare to the baseline (say, A2A), when the exogenous variables are non-monotonic? Would you discretise the interval of potential values they can assume and assume monotonicity?

Q4. Why are React and many of the papers that followed not mentioned or discussed in the Methodology?

---

### Official Review · Reviewer_cJZy · 2025-11-02

**Soundness:** 2
**Presentation:** 2
**Contribution:** 2
**Rating:** 2
**Confidence:** 4

**Summary:**

The paper introduces a communication protocol that incentivizes coordination among a network of heterogeneous agents. The proposed method separates the coordination problem from the agent's internal cognition, explicitly addressing this challenge as part of the "external" communication protocol. The proposed Ripple Effect Protocol (REP) enables agents to share information about their internal motivations by providing predictions for alternative scenarios (sensitivities). It also includes an aggregation mechanism, built upon the communication framework, that combines the sensitivities of neighboring agents to update internal coordination variables, which are then used in subsequent cognitive steps.

**Strengths:**

1. The paper tackles an important challenge, particularly given the growing interest in developing large networks of intelligent agents. The authors present this challenge convincingly and clearly define the specific problem their work seeks to address within this broader context.

2. The proposed solution is intuitive, easy to understand, and powerful, showing clear benefits in the limited set of experiments presented by the authors.

**Weaknesses:**

1. Secondary aggregation step. The proposed solution seeks to enforce coordination through an external mechanism that aggregates sensitivities from neighboring agents and further enforces consensus via a secondary median-based aggregation step. To my understanding, this second aggregation step is optionally applied through human intervention in tasks requiring stronger coordination. However, this enforced alignment appears to be a workaround; a framework that genuinely incentivizes coordination should enable agents to reach consensus autonomously when appropriate, rather than relying on additional human-controlled mechanisms to impose it. The authors do not provide an ablation study in their experiments to understand the impact of this step.

2. RepClient interface description. The main practical contribution of this work is the RepClient, which the authors plan to release publicly in the future. However, its current description is minimal, limited to only two brief examples of function calls. I strongly encourage the authors to provide a more detailed explanation of the client’s structure and internal processes, including a clear template or guideline for the LLM prompts required to effectively use the client and reproduce the intended results in scenarios beyond the three experiments presented in the paper. Furthermore, a step-by-step visualization of the client’s workflow would greatly enhance clarity and help readers form a comprehensive understanding of the contribution. Without these additions, it is difficult to assess the true value of the work, which currently appears rather limited.

3. Experimental setup. While the authors employ three distinct settings to demonstrate the effectiveness of their method, they only compare it against a single baseline: the A2A protocol. This severely limited comparison makes it difficult to assess the realistic benefits of the proposed approach, particularly since the API calls for the two methods appear to differ significantly. Specifically, the REPL’s calls include sentences that explicitly encourage coordination (e.g., “Identify alliance patterns” and “Balance individual optimization with collective sustainability”), which are absent from the baseline’s prompts. This discrepancy raises concerns about the fairness of the comparison, as one method is inherently more biased toward cooperative behavior. I encourage the authors to conduct a more rigorous evaluation, including comparisons with additional baselines, to truly gauge the performance benefits. Moreover, the proposed communication protocol involves multiple sequential steps, yet the relative contribution of each step remains unclear. I recommend performing an ablation study to better quantify the importance of individual components and to clarify which aspects of the framework are most critical to its performance.

4. Repetitions. While reading the paper, I noticed a number of identical or semantically equivalent repetitions throughout the text. For instance, the textual example in lines 173–174 is almost identically repeated in lines 306–307. This is just one example; similar repetitions appear in the descriptions of the domains. I encourage the authors to remove such redundancies, which significantly impact the readability of the manuscript. Doing so would also create space to incorporate the additional details about their approach suggested above.

**Questions:**

See weaknesses summarized above.

---

### Meta-Review · Area_Chair_wJgc · 2025-12-16

**Summary:**

This paper proposes the Ripple Effect Protocol (REP), motivated by the observation that existing agent-to-agent protocols (such as A2A/ACP) primarily focus on "communication" and lack a robust framework for handling "coordination" at the protocol layer. The authors claim that by having each agent share "sensitivity" (reactions to counterfactuals) in addition to decisions, and by updating coordination variables through local neighborhood aggregation (with an optional global consensus step), REP demonstrates better coordination and convergence than A2A baselines across three domains: Beer Game, Movie Coordination, and Fishbanks.

However, the reviews are mixed, with concerns centering on: (1) fairness of baseline comparisons and insufficient ablation studies; (2) lack of clarity regarding the protocol's positioning and description; (3) idealized assumptions regarding uncertainty estimation and associated computational/communication overheads; and (4) the strength of the models used (SOTA/open weights) and the validity of result interpretations.

**The authors did not submit author comments during the discussion period.**

**Reviewer Concerns:**

* **Fairness of Comparison & Insufficient Ablation:** It is not clearly distinguished whether REP's improvement stems from the "structured aggregation/update" itself or simply from differences in prompts or information quantity (cJZy, kJcU). Ablation studies demonstrating the individual contributions of updates, aggregation, and the two-stage process (global consensus) are required (cJZy, kJcU).
* **Unclear Protocol Positioning/Description:** The paper needs to clarify from the outset how this differs fundamentally from mere message extensions of existing protocols—specifically highlighting that it is a "layer" possessing state, broadcast/aggregation mechanisms, and update rules (kJcU).
* **Idealized Uncertainty (Sensitivity) Assumptions & Costs:** There is a lack of evaluation regarding the reliability of sensitivity estimation, its validity in non-monotonic, asynchronous, and (potentially) non-cooperative/dishonest environments, as well as the calculation and communication overheads involved (iupu).
* **Reproducibility/Implementation Details:** The internal workflows and prompt specifications for the REPClient/SDK are only outlined in broad terms, which is insufficient for reproducibility (cJZy).
* **Experimental Validity & Strength of Claims:** Validation using stronger/open-weight models, deeper interpretation of the Fishbanks results (whether it merely "delays collapse"), and qualitative analysis of the sensitivity messages are lacking (6Pwk).

**Reviewer Scores:**

**The authors did not submit author comments during the discussion period.** Therefore, it is reasonable to assume that, at best, all reviewers will maintain their scores (while scores may drop, they are unlikely to rise).

* Reviewer cJZy: **2 → 2**
* Reviewer iupu: **4 → 4**
* Reviewer 6Pwk: **8 → 8**
* Reviewer kJcU: **4 → 4**

Predicted Average: (2 + 4 + 8 + 4) / 4 = **4.5**

---

### Decision · Program_Chairs · 2026-01-26

Reject